# The Effect of Time Monitoring on the Development of Time-Based Prospective Memory among Children Aged 7–11 Years Old

**DOI:** 10.3390/bs14030233

**Published:** 2024-03-13

**Authors:** Yinya Wang, Zhi Ren, Yaqi Yue, Xi Zheng, Xinyuan Zhang, Lijuan Wang

**Affiliations:** 1School of Psychology, Northeast Normal University, No. 5268 Renmin Street, Changchun 130024, China; wangyy022@nenu.edu.cn (Y.W.); yueyq291@nenu.edu.cn (Y.Y.); zhengqian@nenu.edu.cn (X.Z.); zhangxy878@nenu.edu.cn (X.Z.); 2Department and Institute of Psychology, Ningbo University, No. 818 Fenghua Road, Ningbo 315211, China; renzhi@nbu.edu.cn

**Keywords:** school-age children, time-based prospective memory, time monitoring, time monitoring frequency

## Abstract

Time-based prospective memory (TBPM) refers to the ability of an individual to successfully execute an expected plan in the future at a certain time point or after a definite period of time. This study investigated the potential developmental mechanism of school-age children’s TBPM ability from the perspective of time monitoring. Experiment 1 used a between-subjects design of three ages (7, 9, 11) × two TBPM types (time point, time period) to investigate the trends and characteristics of two types of TBPM in children aged 7–11 years old. Experiment 2 used a between-subjects design of three ages (7, 9, 11) × two monitoring conditions (free monitoring, fixed monitoring) to investigate differences in two types of TBPM and monitoring behavior among school-age children under different monitoring conditions. These results showed that the age effect of TBPM was affected by the type of prospective memory (PM) and that time-point PM performance was significantly better than time-period PM performance among school-age children. These findings indicate that clear and definite external cues are helpful for school-age children in performing TBPM tasks. Moreover, there were significant differences found in the TBPM performance of school-age children under different time monitoring conditions. The performance of older children was significantly better than that of younger children. This indicates that older children can better allocate their attentional resources and use time monitoring strategies to improve their performance in PM tasks. Accordingly, this study showed that the TBPM ability of 7- to 11-year-old children is in continuous development and that the time monitoring behavior in the TBPM task is affected by task type and monitoring conditions.

## 1. Introduction

Prospective memory (PM) refers to an individual’s ability to carry out future intentions [1,2]. PM is a basic skill for an individual to achieve and maintain an independent life and is related to one’s future development and the implementation of one’s plans. PM is divided according to cue characteristics into event-based prospective memory (EBPM) and time-based prospective memory (TBPM). EBPM refers to the execution of a plan or intention in the context of a specific cue in the future, such as buying a pencil when passing by a convenience store on the way home from work. TBPM refers to the execution of a plan or intention by an individual at a certain time point in the future or after a defined period of time, such as remembering to call a classmate at 4 p.m. or taking a cake out of the oven within 30 min. EBPM has obvious external cues, while TBPM only uses a certain time point or period of time as cues, and these cues are more concealed [3]. To complete a TBPM task, individuals need to activate both internal and external monitoring methods; that is, a target task is achieved through a combination of accurate assessment of the target time and external monitoring behavior. Therefore, TBPM is a more typical PM type.

According to the multiprocess view (MPV), a TBPM task may involve more control processes that require more cognitive resource support [4]. Preparatory attention and memory processes theory (PAM)further notes that the successful execution of PM requires attentional resources. When an individual needs to perform a PM task, even if the cues of PM have not yet appeared, the individual can always be in a readiness state to monitor the environment for the appearance of target cues, and this monitoring is a demanding process that taxes cognitive resources [5,6,7,8,9]. Therefore, the key to elucidating the process characteristics of PM is to understand how individuals complete TBPM by allocating cognitive resources.

To date, only four studies have focused on the development of TBPM in school children [10,11,12,13], and the results are mixed. Ceci and Bronfenbrenner (1985) [10] first used the family scenario simulation method to investigate the development of TBPM in children aged 10–14 years old. Their PM task entailed examining whether a child was able to remove a cake from the oven after 45 min or to stop charging a battery after 30 min. Their ongoing task (OT) was a video game, where the children could monitor the lapsed time by means of a clock check. These results indicated a significant age effect in PM performance, i.e., the PM performance of the older children was significantly better than that of the younger children. Unfortunately, this study did not report the OT performance of the children. Subsequently, Kerns (2000) [11] developed a new laboratory TBPM task, “CyberCruiser” (a computer game program). In this game, 6- to 10-year-old children were required to complete a TBPM task, i.e., the participants were asked to stop playing the computer game and refuel a car whose fuel gauge, which was depleted at a steady rate, was either continuously displayed on the screen or hidden [14]. These results showed a significant age effect on TBPM performance [12,14,15,16]. However, some studies have reached different conclusions. Nigro et al. (2002) [17] studied the TBPM performance of 7- to 11-year-old children. These subjects were asked to complete a 15 min math and puzzle game (OT) and at the same time to remember to do something at a specific time; no significant age effect was found. Mäntylä et al. (2007) [18] asked 8- to 12-year-old children to press a button every 5 min (TBPM task) while watching a video (OT), and these participants could check the clock by pressing another button. These results were the same, i.e., no significant age effect was found. Therefore, there are still inconsistent conclusions about whether school-age children’s TBPM ability increases with age. The causes of these inconsistent conclusions might be differences in the experimental procedures and methods used to manipulate the monitored variables. In previous studies, the amount of time monitoring when participants performed the focal TBPM task was not limited. Therefore, the age effect may not be significant when the PM task is not very difficult. Thus, the purpose of this study was to use the classic “Dual Tasks Paradigm” proposed by Einstein and McDaniel (1990) [2] to control these monitoring conditions and to systematically investigate the development and intrinsic driving mechanisms of school-age children’s TBPM ability.

Time monitoring behavior is the core element in the study of TBPM [10,19,20,21]. The execution of a TBPM task requires the monitoring of an external clock. Monitoring can help individuals effectively deploy attentional resources so that they can implement their intentions at the appropriate time [21]. Studies have confirmed that TBPM is associated with time monitoring behavior and that time monitoring [14,18,22,23,24,25,26], behavior is an important factor in the performance of TBPM [19,21,27]. Individuals adopt different monitoring frequencies and monitoring strategies based on task demands. Under an unrestricted time-monitoring condition, individuals can rely on both internal and external attention to process time information, whereas with a restricted condition, individuals are unable to rely sufficiently on external information for feedback and must rely more on internal attention [28]. Studies usually use one of two methods to evaluate the effect of time monitoring. One method is to measure time monitoring according to absolute time monitoring frequency, that is, the total number of clock checks that subjects perform in the entire PM task. Absolute time monitoring frequency can positively predict an individual’s TBPM performance; that is, the higher the monitoring frequency is, the better the PM performance [27,29,30]. The other method uses the relative time monitoring frequency, i.e., the distribution of the distance between the time point of each recorded clock check by the subject and the target time; this indicator of time monitoring is more concerned with the distance relationship between the location (time point) used by the individual in time monitoring and the target time of TBPM, thus emphasizing the tactical nature of clock checks [29]. Most studies also only use the absolute frequency of time monitoring as an important indicator and this rough measurement may also be one of the reasons for the mixed results in the literature. Compared to absolute time monitoring, the indicator of relative time monitoring is more refined and sensitive to the cost of TBPM monitoring [31]. Relative time monitoring usually measures the relationship between the number of clock checks by the participants in the last time interval (right before the target time) and PM performance. These results show that the more times a participant checks the clock in the last period, the better their TBPM performance is [24,29,30,32]. Maylor et al. (2002) [33] have analyzed successful trials of TBPM tasks and reported that the frequency of clock monitoring increases as the target time approaches; however, this result was not observed in the failure trials [33]. Therefore, examining strategic clock checking may be a more effective method for enhancing an individual’s TBPM. For school-age children, as their time-monitoring ability develops, they may rely on different attention methods to process time information under different time-monitoring conditions. Hence, in this way, the potential processing mechanism of TBPM can be better understood. This study comprehensively considered two monitoring methods and compared the effect of time monitoring on the development of TBPM among school-age children.

The TBPM performance of school-age children is also affected by task type. TBPM can be further divided into time-point PM and time-period PM. If the TBPM time cue is a definite time point, then the time information it provides is relatively clear, and children can accurately predict when this TBPM cue will appear. This means they can flexibly and efficiently allocate attentional resources based on explicit time information and activate target cues within the relevant time window, achieving the spontaneous retrieval of PM intentions [34,35,36]. If the TBPM time cue is a time range, then the time information it provides is rather vague, i.e., it provides relatively few or no environmental cues, and it cannot support the spontaneous retrieval and processing of intention [37]. In other words, under relatively fuzzy time cues, individuals may be less effective at allocating attentional resources, thus affecting their execution of TBPM tasks [38]. This distinction is similar to that between focused event cues (cues related to cognitive processing involved in the task in progress) and non-focused event cues (cues unrelated to cognitive processing involved in the task in progress) in event-based prospective memory. The focused event cue task involves a relatively spontaneous extraction process and is less affected by age [39]; non-focused event cues involve relatively difficult, strategic cognitive processes that are strongly influenced by age [40]. However, thus far, only Haines et al. (2020) [41] have compared the differences in performance between young and elderly people in two TBPM tasks, and their results revealed no significant age-related differences between these two types of TBPM [41]. The cognitive abilities, such as executive function, of school-age children gradually increase with age, while their ability to mobilize attentional resources also gradually improves. By controlling for the type of TBPM task, we can provide evidence for the developmental characteristics of TBPM among school-age children.

In summary, to better understand the relationship between external time monitoring and prospective memory task performance, we will review previous studies (see Table 1 for details). According to previous research, this study aimed to further understand the developmental characteristics of TBPM ability and the time monitoring strategies and mechanisms of school-age children by manipulating TBPM type and time monitoring conditions. This study hypothesized that the performance of time-based prospective memory of school-age children improves with age, but the development of two types of time-based prospective memory abilities is not synchronous; the performance of time-period prospective memory is better than that of time point prospective memory. Under the two monitoring conditions, the performance of prospective memory of school-age children shows significant age-related changes, and the performance of older children is better.

## 2. Experiment 1: The Development of Different Types of TBPM in Children Aged 7–11 Years

### 2.1. Participants

G*Power was used to calculate the sample size. Since the effect size *f* = 0.4, *α* = 0.05, and 1 − *β* = 0.95, the total needed sample size was calculated to be 100. Experiment 1 recruited 120 children, including 61 boys and 59 girls, from an elementary school in Changchun. Among them, 40 children were 7 years old (*M*_age_ = 7.65, *SD*_age_ = 0.69), 40 were 9 years old (*M*_age_ = 9.3, *SD*_age_ = 0.51), and 40 were 11 years old (*M*_age_ = 10.63, *SD*_age_ = 0.62). The physical examination at admission showed that all the participants were healthy, had normal intelligence, had visual acuity or corrected visual acuity of 1.0 or above, had no color blindness, and were all right-handed. The subjects had not participated in similar experiments before.

### 2.2. Experimental Design and Materials

The experiment used a between-subjects design of 3 ages (7, 9, 11) × 2 TBPM types (time-point PM task, time-period PM task) to investigate the development trend and monitoring frequency of different types of TBPM among school-age children. The accuracy of the TBPM task, the accuracy of the OT, and the time monitoring frequency were measured as dependent variables. The experimental material for the ongoing task (OT) was the living task, which consists of 64 pictures, with the aim of determining whether the objects presented in the pictures are alive. For example, apple trees, pine trees, flowers, mushrooms, plants, and rice are considered as animate objects, and batteries, beds, bread, cars, chess pieces, and clocks are considered as inanimate objects. After processing by Photoshop software, the pixel size was 720 × 540. The experimental materials for the PM task were collected with a timing clock (0:00) written in Python.

### 2.3. Experimental Task

#### 2.3.1. Time-Based Prospective Memory Task

Time-point-based PM task: Subjects were told to remember to press the “green” button every 2 min (corresponding to the “Enter” key), (For example, after the prospective memory instructions were introduced, that is, the PM block started timing. The subjects needed to complete the key response within 1 min before and after the target time point. For example, starting from 0:00, the subjects needed to perform the button reaction within 1 min before and after the four time points of 2:00; 4:00; 6:00; 8:00. The scores varied according to the time and location of the key pressed. The closer you were to the target time, the higher your score, and the further away you were, the lower your score). The child was also told that there was a clock in the upper right corner of the interface, which was hidden since the prospective memory task began and was responsible for timing. To remind children to monitor the passage of time, they could view the current clock by pressing the “yellow” button (the computer’s “Ctrl” key); each time the “Ctrl” key was pressed, the clock displayed for two seconds, then was hidden and continued counting. The clock continued to tick regardless of whether the “Enter” key was pressed or not.

Time period-based PM task: the subjects were told to press the “green” key (corresponding to the computer “Enter” key) after every 2 min while performing the task in progress. (For example, after we introduced the prospective memory instruction, that is, the PM block started timing, the subjects needed to make the key response within 1 min after the target time. For example, starting from 0:00, the subjects needed to complete the button reaction within 1 min after 2:00; 4:00; 6:00; 8:00. The scores varied according to the time and location of the key pressed. The closer you were to the target time, the higher your score, and the further away you were, the lower your score). The child was also told that there was a clock in the upper right corner of the interface, which was hidden since the prospective memory task began and was responsible for timing. To remind children to monitor the passage of time, they could view the current clock by pressing the “yellow” button (the computer’s “Ctrl” key); each time the “Ctrl” key was pressed, the clock displayed for two seconds, then was hidden and continued counting. The clock continued to tick regardless of whether the Enter key was pressed or not.

#### 2.3.2. Ongoing Task

A living task, i.e., determining whether there is an animate object in the presented picture. If there was an animate object, the “blue” key should be pressed (corresponding to the “F” key on the computer); if there was an inanimate object, the “red” key should be pressed (corresponding to the “J” key on the computer). The living task included 64 pictures, with the ratio of inanimate and animate being 1:1. In the experiment, each picture was repeatedly presented 3 times. Therefore, there were a total of 192 trials in the OT (96 trials of animate and 96 trials of inanimate). The total time of the formal test was 9 min. During the whole test, the children were required to perform 4 time-point-based PM tasks.

#### 2.3.3. Experimental Procedures

All participants completed the experimental tasks individually in the computer classroom at school. All experimental tasks were programmed in Python. Before the start of the experiment, the children were asked to sit quietly in front of the computer, and the instructions for the time-point PM task test were first displayed on the computer screen. The examiner explained the contents of the instructions to allow the children to fully understand the instructions for the OT, and then the fixation point “+” was shown for 500 ms. Participants were reminded to focus on the appearance of the target stimuli. The OT could be practiced repeatedly until the child could perform correctly according to the experimental requirements. At the end of the practice, the children rested for 3 min and then entered the formal experimental stage. At the end of the formal experiment, the subjects were asked to recall the instructions of the PM task and were given a gift. The whole experimental process lasted 16–20 min. Figure 1 shows the detailed experimental procedure.

### 2.4. Scoring

Time-point-based PM test: The key press reactions 1 min before and after the target time were considered to have correctly performed the time point prospective memory task. However, the score differed depending on the specific time of the key press, with the score being higher when the key was pressed close to the target time. There were four PM tasks, a full score of 12 points. For example, the first target time point was 2:00. The participants would be awarded 1 point for pressing the button between 1′–1′20″ and 2′40″–3′; 2 points for pressing the button between 1′20″–1′40″ and 2′20″–2′40″; and 3 points for pressing the button between 1′40″–2′ and 2′–2′20″. No points were awarded for pressing the button at other time points. The total scores of the four time point test tasks were accumulated as the participants’ performance of time-point-based prospective memory (see Figure 2).

Time-period-based PM test: A button response within 1 min after the target time was considered to indicate correct performance of the PM task. However, the score is different depending on the specific time period of the key press, with the score being higher when the key is pressed close to the target time. There were four PM tasks, a full score of 12 points. For example, the first target time period is: within 1 min after 2:00. The participants will be awarded 1 point for pressing the button between 2′40″ and 3′; 2 points for pressing the button between 2′20″ and 2′40″; and 3 points for pressing the button between 2′ and 2′20″. No points will be awarded for other time points. The total scores of the four time-based prospective memory tasks will be accumulated as the participants’ performance of time-based prospective memory (see Figure 3).

Relative time monitoring frequency: The frequency at which subjects checked the clock within 1 min before the target time point was recorded [24,30,31,44]. In this study, 1 min before the target time point was evenly divided into four intervals (T1, T2, T3, and T4), the number of clock checks performed by the subjects in the T4 interval was subsequently divided by the sum of the clock checks in the interval of T1–T4, and the relative frequencies of the clock checks before the four target time points were summed and divided by 4; that is, the relative clock check frequency = $\left[\frac{\sum_{t=1}^{4}\frac{T4_{t}}{T1_{t}+T2_{t}+T3_{t}+T4_{t}}}{4}\right]\times 100$. This formula was used to obtain the percent of the relative frequency for each participant, and a larger value indicated a greater frequency of strategic time monitoring [31]. OT: the accuracy rate of the ongoing task was recorded; that is, the number of correctly classified animate objects by the subjects was counted.

### 2.5. Results Analysis

The descriptive statistics for PM, OT, and monitoring frequency performance of children aged 7–11 years in different conditions are shown in Table 2.

#### 2.5.1. PM Performance

Then, 3 (ages: 7, 9, 11) × 2 (task types: time-point PM or time-period PM) ANOVA was conducted for the PM performances of the children. The results showed that the main effect of age was significant (*F*_(1, 114)_ = 9.26, *p* < 0.001, *η*^2^ = 0.14), the PM performance of the 11-year-old group (*M* = 8.30, *SD* = 0.51) was significantly better than those of the 9-year-old group (*M* = 6.55, *SD* = 0.52) and the 7-year-old group (*M* = 5.15, *SD* = 0.51); the main effect of TBPM type was significant (*F*_(1, 114)_ = 22.38, *p* < 0.001, *η*^2^ = 0.16). The time-point PM scores (*M* = 8.08, *SD* = 0.42) were significantly better than the time-period PM performance (*M* = 5.25, *SD* = 0.42); there was a significant interaction between age and TBPM type (*F*_(2, 119)_ = 3.07, *p* = 0.05, *η*^2^ = 0.51). Simple effects analysis revealed that for the 7-year-old group (*t* (38) = 3.07, *p* = 0.004) and the 11-year-old group (*t* (38) = 4.67, *p* < 0.001), the time-point PM performance was significantly better than the time-period PM performance. There were no significant differences between the time-point and time-period PM performance in the 9-year-old group (*t* (38) = 0.75, *p* = 0.461). According to the analysis of TBPM development trends, there was no significant difference in the development of time-point PM ability between the 7-year-old group and the 9-year-old group (*p* = 0.99), and for children in the 7-year-old group (*p* = 0.001) and the 9-year-old group (*p* < 0.01), both performances were significantly lower than children in the 11-year-old group. When it comes to the development of point-period PM performance, the performances of the 7-year-old group were significantly lower than those of the 9-year-old group (*p* < 0.05) and 11-year-old groups (*p* < 0.05), and there was no significant difference in the performance between the 9-year-old group and the 11-year-old group (*p* = 1.0). The results show that the development of the two different types of temporal prospective memory ability is not synchronized, and the time period may develop earlier (Figure 4).

#### 2.5.2. OT Performance

Scores of children aged 7–11 years for the OT in the two types of TBPM: A model of 3 (ages: 7, 9, 11) × 2 (TBPM types: time-point PM, time-period PM) was used for the ANOVA of the performance of children aged 7–11 years for the OT in the two types of TBPM; these results showed that the main effect of TBPM type was not significant, *F*_(1, 114)_ = 2.49, *p* > 0.05, and that the main effect of age was significant, *F*
_(1, 114)_ = 10.17, *p* < 0.001, *η*^2^ = 0.15. The performance of the 11-year-old children (*M* = 0.83, *SD* = 0.02) were significantly better than those of the 9-year-old children (*M* = 0.72, *SD* = 0.02) and the 7-year-old children (*M* = 0.71, *SD* = 0.02). The interaction between the two was not significant, *F*_(2, 114)_ = 1.02, *p* > 0.05 (Figure 5).

#### 2.5.3. Monitoring Frequency

Frequency of two types of monitoring for children aged 7–11 years: A model of 3 (ages: 7, 9, 11) × 2(TBPM types: time point, time period) × 2 (frequencies: relative frequency, absolute frequency) was used for the ANOVA of the scores of children under different monitoring frequencies; these results showed that the main effect of age was significant, *F*_(1, 114)_ = 9.04, *p* < 0.001, *η*^2^ = 0.13. Under time-point-based conditions, the monitoring frequency of 11-year-old children (*M* = 0.41, *SD* = 0.02) was significantly greater than that of 9-year-old children (*M* = 0.35, *SD* = 0.02) and 7-year-old children (*M* = 0.25, *SD* = 0.02). Under time-period-based conditions, the monitoring frequency of 11-year-old children (*M* = 0.38, *SD* = 0.02) was significantly greater than that of 9-year-old children (*M* = 0.34, *SD* = 0.02) and 7-year-old children (*M* = 0.24, *SD* = 0.02). The main effect of the monitoring frequency condition was significant, *F* _(1, 114)_ = 5.93, *p* = 0.016, *η*^2^ = 0.05. The relative monitoring frequency (*M* = 0.34, *SD* = 0.01) was significantly greater than the absolute monitoring frequency (*M* = 0.30, *SD* = 0.01). There was a significant interaction between the TBPM type and the monitoring frequency condition, *F* _(1, 114)_ = 11.77, *p* = 0.001, *η*^2^ = 0.09. Simple effects analysis revealed that under the condition of time-point PM, there were significant differences in the scores between the relative monitoring frequency and absolute monitoring frequency (*t* (59) = 4.31, *p* < 0.001) and that the scores for relative frequency (*M* = 0.47, *SD* = 0.02) were better than those for absolute frequency (*M* = 0.43, *SD* = 0.02). Under the time-period PM condition, there was no significant difference in scores between the relative monitoring frequency (*M* = 0.21, *SD* = 0.02) and the absolute monitoring frequency (*M* = 0.22, *SD* = 0.02), *t* (59) = 0.67, *p* = 0.50. These results indicate that the monitoring frequency for the two types of TBPM differ with children’s age and are shown in Figure 6.

Correlation between time monitoring behavior and TBPM/correlation analysis of the TBPM scores and the time monitoring frequency of the children aged 7–11 years revealed that the correlation coefficient between the relative time monitoring frequency and the TBPM score was 0.72 (*p* < 0.001). The higher the relative time monitoring frequency, the better the performance of time-based prospective memory; the correlation coefficient between the absolute time monitoring frequency and the TBPM score was 0.67 (*p* < 0.001). The higher the absolute time monitoring frequency, the better the time-based prospective memory performance. Further, the performance of time-based prospective memory was taken as a dependent variable, and age, temporal prospective memory type, and monitoring frequency type were taken as independent variables for stepwise regression analysis. The results show that relative monitoring frequency has significant regression coefficient on time-based prospective memory, *β* = 0.73, *t* = 11.37, *p* < 0.001; relative monitoring frequency has significant positive predictive effect on time-based prospective memory performance. The margin of the regression coefficient of age on time-based prospective memory is significant, *β =* 0.13, *t =* 1.97, *p =* 0.051 (Table 3).

## 3. Experiment 2: The Development of TBPM in Children Aged 7–11 Years under Different Monitoring Frequencies

As shown in Experiment 1, the time-point PM scores of school-age children were significantly better than the time-period PM scores, with a clear development trend with age. Therefore, in Experiment 2, only the time-point PM task was used to further investigate the influencing factors and mechanisms behind school-age children’s TBPM.

### 3.1. Participants

G*Power was used to calculate the sample size. Since the effect size *f* = 0.4, *α* = 0.05, and 1 − *β* = 0.95, the total required sample size was calculated to be 100. Experiment 2 recruited 181 children, including 88 boys and 93 girls, from an elementary school in Changchun. Among them, 61 children were in the 7-year-old group (*M*_age_ = 7.59, *SD*_age_ = 0.76), 60 children were in the 9-year-old group (*M*_age_ = 9.22, *SD*_age_ = 0.48), and 60 children were in the 11-year-old group (*M*_age_ = 10.65, *SD*_age_ = 0.63). The physical examination at admission showed that all the subjects were healthy, had normal intelligence, had visual acuity or corrected visual acuity of 1.0 or above, had no color blindness, and were all right-handed. All the subjects had not participated in similar experiments.

### 3.2. Experimental Design and Materials

Experiment 2 used a between-subjects design of 3 ages (7, 9, 11) × 2 monitoring conditions (free monitoring, fixed monitoring) to investigate the developmental characteristics of TBPM among school-age children under different monitoring conditions. With reference to Mioni and Stablum (2014) [24], under free monitoring conditions, the number of times the subjects checked their clock was not limited; under fixed monitoring conditions, the number of times the subjects checked their clock was limited (4 times). Their accuracy in the PM task, accuracy in the OT, and monitoring frequency were measured as dependent variables. All materials were the same as in Experiment 1.

### 3.3. Experimental Task

#### 3.3.1. Time-Based Prospective Memory Task

TBPM task under the fixed monitoring condition: the subjects were told to press the “green” button (corresponding to the computer “Enter” button) every 2 min (2:00; 4:00; 6:00; 8:00) while performing the ongoing task. Participants could press the “yellow” button (corresponding to the “Ctrl” key on the computer) to check the clock. Each time the “Ctrl” key was pressed, the clock was displayed for 2 s before it disappeared. The clock was always ticking whether or not the “Ctrl” key was pressed. However, the subjects were required to check the clock only 4 times every 2 min, and the number of times was displayed next to the clock (1–4 times), and the number of times of checking the clock was reset every 2 min. If the clock was checked more than 4 times every two minutes, the clock and the number of checks were no longer presented. TBPM task under the free monitoring condition: the subjects were told to press the “green” button (corresponding to the computer “Enter” button) every 2 min (2:00; 4:00; 6:00; 8:00) while performing the ongoing task. Participants could press the “yellow” key (corresponding to the “Ctrl” key on the computer) to check the clock. Each time the “Ctrl” key was pressed, the clock was displayed for 2 s before it disappeared. The clock was always ticking whether or not the “Ctrl” key was pressed. However, the number of times they looked at the clock was not limited, that is, they could monitor the time at any time as needed.

#### 3.3.2. Ongoing Task

The ongoing task was the same as in Experiment 1.

#### 3.3.3. Experimental Procedure

Each child was tested individually in the computer room at school. The experimental programs used were written in Python. The examiner provided instructions for the OT, and the participants began to practice the OT. At the end of the practice stage, the examiner informed the subjects of the instructions for the PM task, and the formal experimental test began after the subjects had understood the instructions. At the end of the experiment, the subjects were asked to recall the instructions of the PM task and were given a gift. The whole experiment lasted 16–20 min. Figure 1 shows the flow chart of the TBPM experiment.

### 3.4. Scoring Methods

TBPM score in the fixed or free monitoring condition: this was the same as that of the time-point PM score in Experiment 1.

Two indicators of absolute time monitoring frequency and relative time monitoring frequency were used. According to previous studies [24,30,31,44], the calculation of relative time monitoring frequency was the same as that in Experiment 1. The absolute time monitoring frequency was calculated as the total number of clock checks by the subjects in the minute before the four target time points divided by the number of clock checks in the entire PM task [31]. Repeated measures ANOVA was used to compare and thus examine whether age predicted the time monitoring and TBPM performance. In the OT, the number of correctly classified items was counted as the score, and the percentage of subjects correctly classifying the animate objects was recorded.

### 3.5. Results Analysis

The descriptive statistics of the TBPM, OT, and monitoring frequency performance for the children aged 7–11 years under different monitoring indicators are shown in Table 4.

#### 3.5.1. PM Performance

The TBPM performance of children in the 7- to 11-year-old group under the two monitoring conditions: for the test with 3 (ages: 7, 9, 11) × 2 (monitoring conditions: fixed monitoring, free monitoring), the ANOVA showed that the main effect of age was significant, *F*_(2, 175)_ = 17.94, *p* < 0.001, *η*^2^ = 0.17; the PM performance of the 11-year-old group (*M* = 8.98, *SD* = 0.38) was significantly better than that of the 9-year-old group (*M* = 6.87, *SD* = 0.38) and the 7-year-old group (*M* = 5.82, *SD* = 0.38); the main effect of the monitoring condition was not significant, *F*_(1, 175)_ = 0.25, *p =* 0.62; and the interaction between age and monitoring condition was significant, *F* _(2, 175)_ = 4.32, *p* = 0.02, *η*^2^ = 0.05. Further analysis revealed that there was no significant difference in the PM performance of the 7-year-old group between the fixed monitoring condition (*M* = 5.90, *SD* = 0.53) and the free monitoring condition (*M* = 5.73, *SD* = 0.54), *t* (29) = 0.21, *p* = 0.84; there was no significant difference in the PM performance of the 9-year-old group between the fixed monitoring condition (*M =* 7.43, *SD* = 0.54) and the free monitoring condition (*M* = 6.30, *SD* = 0.54), *t* (29) = 1.28, *p* = 0.21; the PM performance of the 11-year-old group under the free monitoring condition (*M* = 9.97, *SD* = 0.54) was significantly better than under the fixed monitoring condition (*M* = 8.00, *SD* = 0.54), *t* (29) = 3.39, *p* = 0.002. These results suggest that under different monitoring conditions, age-related differences appear in the TBPM performance of children in the 7- to 11-year-old group. These results are shown in Figure 7.

#### 3.5.2. OT Performance

The OT performance of the children in the 7- to 11-year-old group under the two monitoring conditions: for the test with 3 (ages: 7, 9, 11) × 2 (monitoring conditions: fixed monitoring, free monitoring), the ANOVA showed that the main effect of age was not significant, *F* _(2, 175)_ = 0.06, *p* = 0.001; the OT performance of the 11-year-old group was not significantly different from those of the 9-year-old group (*M* = 0.74, *SD* = 0.01) and the 7-year-old group (*M* = 0.75, *SD* = 0.01); the main effect of the monitoring condition was not significant; the OT performance under the fixed monitor condition (*M* = 0.75, *SD* = 0.01) was not significantly different from those under the free monitoring condition (*M* = 0.74, *SD* = 0.01), *F*_(1, 175)_ = 0.07, *p* = 0.797; the interaction between age and the monitoring condition was not significant, *F* _(2, 175)_ = 0.01, *p* = 0.997. These results suggest that under different monitoring conditions, there are no significant age-related differences in the OT performance of children in the 7- to 11-year-old group. Results are shown in Figure 8.

#### 3.5.3. Monitoring Frequency

Time monitoring frequency: A model of 3 (ages: 7, 9, 11 years) × 2 monitoring conditions (fixed, free) × 2 frequencies (relative frequency, absolute frequency) was used for the ANOVA; these results show that the main effect of age was significant, *F* _(2, 175)_ = 23.64, *p* < 0.001, *η*^2^ = 0.21; the monitoring frequency scores of the 11-year-old group (*M* = 0.47, *SD* = 0.01) were significantly greater than those of the 9-year-old group (*M =* 0.32, *SD* = 0.01) and the 7-year-old group (*M* = 0.32, *SD* = 0.01); the main effect of monitoring frequency was significant, *F*_(1, 175)_ = 90.59, *p* < 0.001, *η*^2^ = 0.34. The interaction of age and monitoring frequency was significant, *F*_(2, 175)_ = 9.95, *p* < 0.001, *η*^2^ = 0.10. For 7-year-old children, the absolute frequency (*M* = 0.43, *SD* = 0.02) was significantly greater than the relative frequency (*M* = 0.22, *SD* = 0.02), *t*(59) = 7.45, *p* < 0.001; for 9-year-old children, the absolute frequency (*M* = 0.42, *SD* = 0.02) was also significantly greater than the relative frequency (*M* = 0.23, *SD* = 0.02), *t*(59) = 6.53, *p* < 0.001; however, for the 11-year-old group, the difference between relative frequency (*M* = 0.44, *SD* = 0.02) and absolute frequency (*M* = 0.49 *SD* = 0.02) was not significant, *t*(59) = 1.78, *p* = 0.08. There was a significant interaction between age, monitoring conditions, and monitoring frequency *F*_(2, 175)_ = 9.54, *p* < 0.001, *η*^2^ = 0.09. Further analysis revealed that, for 7-year-olds, under the fixed monitoring conditions, the absolute frequency (*M* = 0.48, *SD* = 0.17) was also significantly greater than the relative frequency (*M* = 0.21, *SD* = 0.16); under the free monitoring conditions, the absolute frequency (*M* = 0.39, *SD* = 0.20) was also significantly greater than the relative frequency (*M* = 0.22, *SD* = 0.14); for 9-year-olds, under the fixed monitoring conditions, the absolute frequency (*M* = 0.44, *SD* = 0.20) was also significantly greater than the relative frequency (*M* = 0.25, *SD* = 0.14); under the free monitoring conditions, the absolute frequency (*M* = 0.57, *SD* = 0.15) was also significantly greater than the relative frequency (*M* = 0.40, *SD* = 0.16); for 11-year-olds, under the fixed monitoring conditions, the relative frequency (*M* = 0.49, *SD* = 0.21) was more than the absolute frequency (*M* = 0.42, *SD* = 0.20); under the fixed monitoring conditions, the absolute frequency (*M* = 0.57, *SD* = 0.15) than the relative frequency (*M* = 0.40, *SD* = 0.16). The results showed that the younger children (7 and 9 years old) had poor monitoring strategy and could only improve prospective memory performance by frequently looking at the clock; older (11-year-old) children were better able to use time monitoring strategies when there was a limit on the number of monitoring sessions. The results are shown in Figure 9.

Correlation between time monitoring behavior and TBPM: Correlation analysis of the TBPM scores and the time monitoring frequency of the children aged 7–11 years revealed that the correlation coefficient between the relative time monitoring frequency and the TBPM score was 0.47 (*p* < 0.001). The higher the relative time monitoring frequency, the better the time-based prospective memory performance. Further, the performance of temporal prospective memory was taken as the dependent variable, and age, monitoring conditions, and monitoring frequency type were taken as independent variables for stepwise regression analysis. The results showed that the regression coefficient of age on time-based prospective memory was significant, *β =* 0.40, *t =* 5.78, *p <* 0.001. Age therefore has a significant positive predictive effect on time-based prospective memory performance (Table 5).

## 4. Discussion

### 4.1. The Development Trend of TBPM among School-Age Children

First, the primary purpose of this study was to evaluate the developmental characteristics of TBPM among school-age children aged 7–11 years. The results of this study show that older school-age children (11 years old) performed better on the TBPM task and the OT. Thus, the TBPM ability of school-age children is still in a state of continuous development. These results are consistent with previous studies [11,12,14,15,16,45] and are in line with the inverted “U” shaped development curve of PM [40,46,47,48,49,50,51]. Previous studies have shown that compared to an EBPM task, children’s ability to successfully complete a TBPM task develops relatively late [45,48]. In addition, a TBPM task has a greater impact on older children than on younger children [17,52]. This study has confirmed this point of view, indicating that elementary school may be an important stage in the development of TBPM ability. This study has also revealed no significant differences in the scores between the 9-year-old group and the 7-year-old group [17,53]. This indicates that the critical development stage of TBPM among school-age children may be 9 years of age. In addition, from the perspective of cognitive development, PM and executive function increase with age [45,54,55,56,57]. The age effect of PM may be the result of differences in the focal children’s executive functions, a topic worthy of further study.

Second, this study shows that the performance of time-point prospective memory of school-age children is significantly better than that of time-period prospective memory. In addition, the performance of time-point and time-period prospective memory of school-age children continues to develop with age. This shows that clear external time cues are more helpful in improving the performance of the time-point prospective memory of school-age children, which verifies the previous research results [58,59]. At the same time, the two types of prospective memory abilities of school-age children do not develop in sync, and the time-period prospective memory ability mainly concentrates on the stage of 7- to 9-year-old children. The time-point prospective memory ability is mainly concentrated in the stage of 9- to 11-year-old children. The development of time-period prospective memory takes precedence over the development of time-point prospective memory. This pattern of development may be due to the availability of attention resources. As noted earlier, time-point and time period-prospective memory tasks require different attentional resources. Time-point prospective memory is a clear time point, and as mentioned earlier, time-point and time-period prospective memory tasks have different requirements for attention resources. Time-point prospective memory is a clear time point, providing relatively clear time information, children need to flexibly and effectively allocate attention resources, activate target cues within the associated time window, so as to achieve spontaneous extraction of prospective memory intentions [34,35,36]. Compared to time-period PM, the internal attention given to time-point PM is more focused on approaching the target time. Because the emergence timing of target time in time-point PM tasks is more explicit, school-age children may be more inclined to adopt non-conservative information processing when the context in which the external cues emerge is very clear. When they are far from their target time point, school-age children are less likely to actively check the time to obtain feedback [28,60,61]. Moreover, children may estimate a definite time interval (e.g., 2 min) rather than a vague time interval (e.g., 2 min later) when the time information is clearer. This allows school-age children to allocate attentional resources more explicitly and flexibly. From the perspective of attentional resource processing, the closer it gets to the target time, the more attentional resources the participants invest, resulting in higher attention effectiveness [62]. According to this view, the effectiveness of internal attention in the time-point PM task is significantly higher than in the time-period PM task. However, it is worth noting that when participants are engaged in prospective memory tasks with time cues, external attention is often more reliable than internal attention, leading individuals to rely more on external attention to process temporal information [28]. In terms of external attention, individuals can both monitor the passage of time and search for external target cues. The processing of temporal information typically requires a significant amount of self-initiated attentional resources [19,63], whereas individuals need less attentional resources to process explicit external cues, which can lead to better prospective memory performance [64,65]. Therefore, the development of prospective memory is later than that of temporal prospective memory at time points, which may be because children’s attentional resources are closely related to executive function development, and older children’s attentional resources and executive function development are better. However, it should be pointed out that this study did not measure executive function, and this inference needs to be confirmed by further research examining the effect of executive function.

### 4.2. The Role of Time Monitoring in the TBPM Development of School-Age Children

Different from previous studies, this study examined the performance of school-age children in two types of temporal prospective memory and two types of monitoring conditions, and found that the performance of the older children (11-year-old group) was better than that of the younger children (7-year-old group and 9-year-old group). In addition, this study simultaneously used two methods to evaluate the impact of time monitoring behavior on the development of temporal prospective memory in school-age children. This study found that school children aged 7–11 years showed different time monitoring behaviors in the process of completing temporal prospective memory tasks. Specifically, when school-age children completed different types of temporal prospective memory tasks, it was found that the relative monitoring frequency had a significant positive predictive effect on temporal prospective memory performance. Age was marginal to prospective memory performance. In other words, the better the time monitoring strategy of school-age children, the better the performance of temporal prospective memory. At the same time, under the condition of prospective memory at time point, the relative monitoring frequency of school-age children was more than the absolute monitoring frequency. These results indicate that when the time cue is more clear, the time monitoring strategy of school-age children can improve the performance of temporal prospective memory. When school-age children completed temporal prospective memory tasks under different monitoring conditions, we found that only age had a positive predictive effect on temporal prospective memory performance. This indicates that older school-age children have higher execution and attention resources and are able to allocate more attention resources to clock monitoring, while younger school-age children may be affected by the number of clock monitoring limits. However, further analysis of variance found that younger children (7- and 9-year-olds) were monitored more frequently in absolute terms than in relative terms in both monitoring conditions. The relative monitoring frequency of older children (11 years old) was higher than the absolute monitoring frequency under fixed monitoring conditions and the absolute monitoring frequency was higher than the relative monitoring frequency under free monitoring conditions. This shows that the monitoring strategy of young children is not good, and only by looking at the clock more times can they ensure the correct completion of temporal prospective memory tasks. For older children, especially when clock monitoring is limited, prospective memory performance can be improved through the strategic use of attention resources due to their better executive and attention resources. Therefore, these results show that time monitoring behavior and strategic time monitoring have important influences on school-age children’s TBPM performance. In addition, for school-age children, external attention is often more reliable than internal attention when processing PM tasks with time cues; thus, they may be more inclined to rely on external attention when processing time information [28]. In summary, as mentioned previously, the development of TBPM among school-age children increases according to and is closely related to their time monitoring behavior [11,12,14,66,67]. In conclusion, this study provides an empirical basis for investigating the effects of children’s time monitoring behavior on prospective memory performance.

Theoretically, the processes of PM involve maintaining intentions in working memory, monitoring a clock for searching target cues, switching task goals, and retrieving or executing intentions. These processes may be challenging for a school-age child whose brain is still developing. Although PAM and dual process theory agree on the basic components involved in PM, they differ in their assumptions on whether attentional control is compulsive. PAM proposes that preparatory attention and memory processes are necessary for successful PM retrieval. On the other hand, dual process theory hypothesizes that in some cases, attention control may be an automatic process. Therefore, the retrieval of time cues may be considered more automatic and less strategic. The assumptions of dual process theory may be reasonable for older children and, thus, have some explanatory meaning. The results of this study supported PAM [5]. Monitoring the environment for the appearance of time cues led to an increase in the cost of the OT. Low TBPM and OT performance was observed in the younger school-age children due to their poor cognitive ability in terms of processing PM cues and retrieval intentions and their inadequate strategies for allocating resources [52]. According to attentional resource allocation theory [68], individual attentional resources can be allocated to different tasks at the same time, while these attentional resources are very limited in nature. In the TBPM task, when the OT occupied too many attentional resources, the resources allocated to the PM task could be reduced, adversely affecting PM performance.

## 5. Conclusions

This study investigated the developmental characteristics and processing mechanisms of school-age children’s TBPM from the perspective of time monitoring and reached the following conclusions:

First, the time-based prospective memory ability of school-age children continues to develop with the increase in age. But the performance of the two types of prospective memory does not develop in sync, the development of school-age children’s prospective memory ability of time point is later than that of time period. The development of time-point prospective memory ability mainly occurs in the 9–11 years old group. The development of time-period prospective memory ability mainly occurs in the 7- to 9-year-old group.

Second, the time monitoring behavior of school-age children showed different age trends under different types of prospective memory tasks, and the time monitoring strategy of older children was better than that of younger children; in particular, when there are clear time cues, children can improve prospective memory performance through strategic monitoring behavior.

In addition, the time monitoring behavior of school-age children showed different age trends under different monitoring conditions. Among them, the monitoring strategy of young children was not good, and the performance of time prospective memory could only be improved by checking the clock frequently. Older children were better able to use time monitoring strategies when there was a limit on the number of monitoring sessions.

## Figures and Tables

**Figure 1 behavsci-14-00233-f001:**
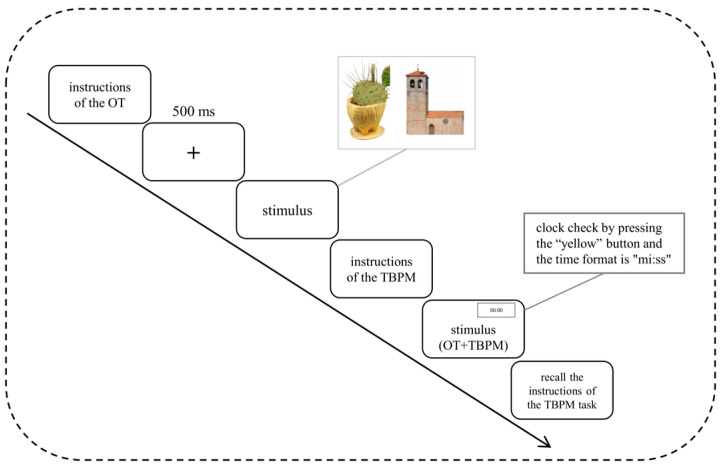
Flowchart of the TBPM task.

**Figure 2 behavsci-14-00233-f002:**
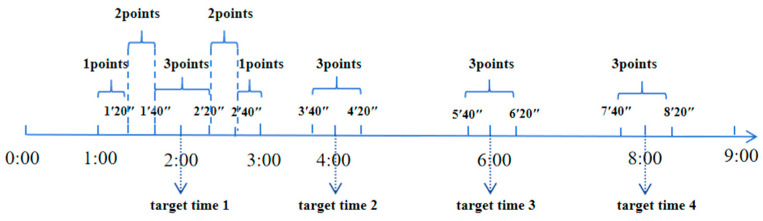
Schematic diagram of time-point prospective memory task scoring.

**Figure 3 behavsci-14-00233-f003:**
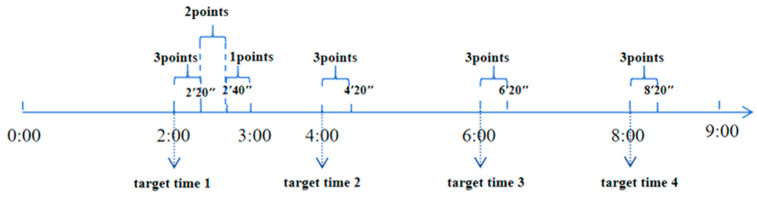
Schematic diagram of time-period prospective memory task scoring.

**Figure 4 behavsci-14-00233-f004:**
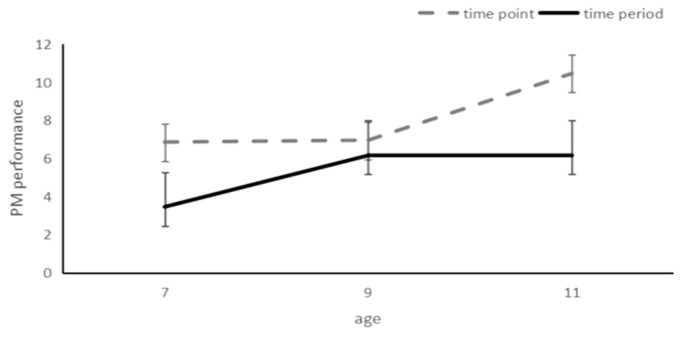
Cores for two types of TBPM for children aged 7–11 years.

**Figure 5 behavsci-14-00233-f005:**
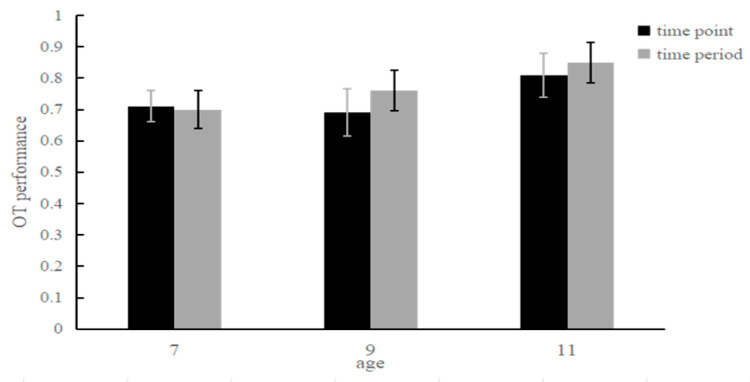
Scores for the OT in the two types of TBPM for children aged 7–11 years.

**Figure 6 behavsci-14-00233-f006:**
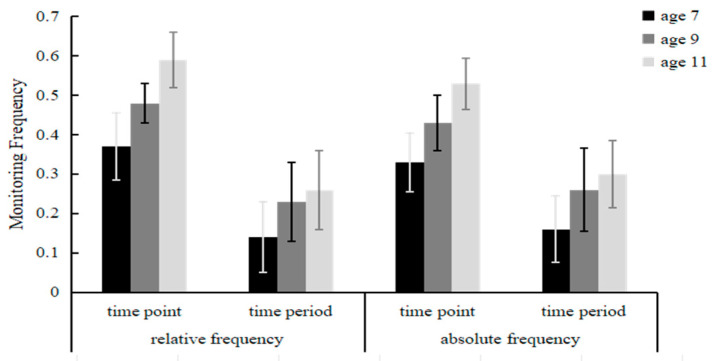
Monitoring frequency for two types of TBPM for children aged 7–11 years.

**Figure 7 behavsci-14-00233-f007:**
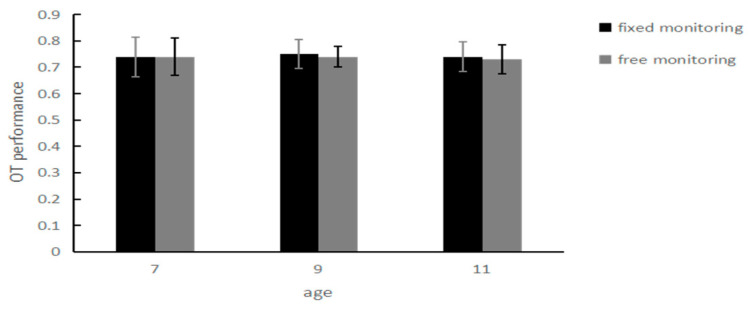
TBPM performance of children aged 7–11 years under different monitoring conditions.

**Figure 8 behavsci-14-00233-f008:**
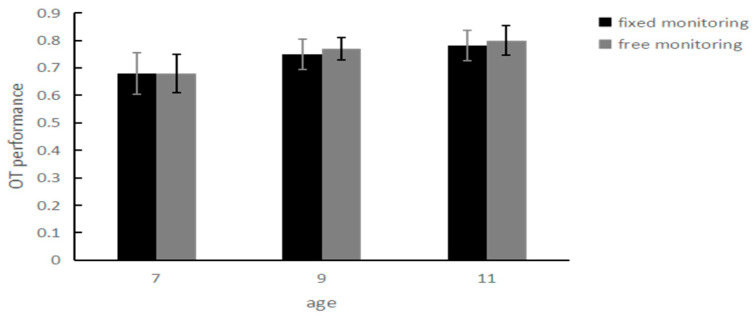
OT scores of children aged 7–11 years under the two monitoring conditions.

**Figure 9 behavsci-14-00233-f009:**
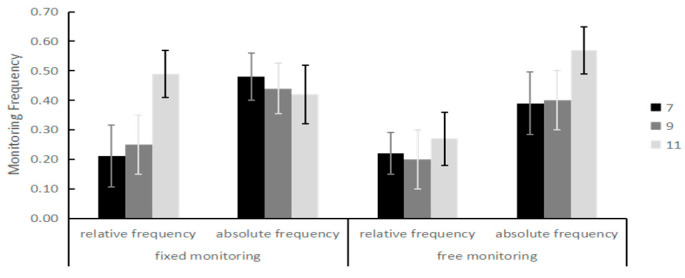
Scores of children aged 7–11 years with different monitoring frequencies.

**Table 1 behavsci-14-00233-t001:** Analysis and results summary of PM time monitoring development research.

Authors, Year	Age	TBPM Task
Ceci and Bronfenbrenner, 1985 [10]	10–14	Remove the cake after 45 min.
Voige et al., 2011 [14]	6–10	Fill up the car with gasoline.
Nigro et al., 2002 [17]	7–11	Do something at a certain time.
Mäntylä et al., 2007 [18]	8–12	Press a key every 5 min;
Mioni and Stablum, 2014 [24]	19–34, 60–88	the key response every five minutes (two monitoring conditions).
Mioni et al., 2020 [30]	22,74	Virtual Week (Rendell and Craik, 2000) [36].
Joly-Burra et al., 2022 [31]	19–86	Each minute button.
Guo and Huang, 2019 [42]	18–24	Press a key every minute.
Haines et al., 2020 [41]	19–30, 69–86	Virtual Week.
Schmidt et al., 2023 [43]	62–85	Press a key every 2 min.
**Ongoing Task**	**Monitoring Measures**	**Relation with PM Performance**
video game	TCC	TCC
electronic driving game	LIC	LIC
jigsaw puzzle	TCE	No report
Watch the video	TCC and LIC	TCC: no; LIC: yes
Watch a movie	TCC, FCC, Index of strategic monitoring	LIC: yes
Virtual Week	TCC and FCC	TCC: yes; LIC: yes
2-back	TCC, FCC, relative and absolute frequency	TCC: yes; relative frequency: yes
1-back	TCC, Time difference indicator	Not reported
Virtual Week	relative and absolute frequency	Not reported
1-back	TCC and FCC	TCC: yes

Abbreviations: FCC, frequency of clock checks for each subinterval of the PM period (usually 1 min); J-shaped graph, this study reports a graphical representation of the relation between the number of clock checks and the passage of time; LIC, number of clock checks in the last subinterval of the PM period (usually 1 min or 30 s); TCC, total number of clock checks.

**Table 2 behavsci-14-00233-t002:** PM, OT, and monitoring frequency performance of children aged 7–11 years (*M* ± *SD*).

	7 Years Old		9 Years Old		11 Years Old	
	*M*	*SD*	*M*	*SD*	*M*	*SD*
Time-point PM	6.85	3.34	6.95	3.65	10.45	1.93
Time-period PM	3.45	3.64	6.15	3.12	6.15	3.65
Time-point OT	0.71	0.11	0.69	0.14	0.81	0.11
Time-period OT	0.70	0.13	0.76	0.17	0.86	0.98
Time-point relative frequency	0.37	0.17	0.48	0.10	0.59	0.14
Time-period relative frequency	0.14	0.18	0.23	0.20	0.26	0.20
Time-point absolute frequency	0.33	0.15	0.43	0.14	0.53	0.13
Time-period absolute frequency	0.16	0.17	0.26	0.21	0.30	0.17

Note: PM: prospective memory; OT: ongoing task.

**Table 3 behavsci-14-00233-t003:** Regression relationships among age, TBPM types, relative frequency, absolute frequency, and TBPM performance.

	Predictor	*β*	*p*	*t*
TBPM	age	0.130	0.051	1.974
	TBPM types	0.092	0.247	1.164
	relative frequency	0.723 ***	<0.001	11.365
	absolute frequency	0.111	0.457	0.756

Note: *** *p* < 0.001.

**Table 4 behavsci-14-00233-t004:** TBPM, OT, and monitoring frequency performance of children aged 7–11 years under different monitoring conditions (*M* ± *SD*).

	7 Years Old		9 Years Old		11 Years Old	
	*M*	*SD*	*M*	*SD*	*M*	*SD*
fixed monitoring TBPM	5.90	2.93	7.43	2.81	8.00	2.42
free monitoring TBPM	5.73	3.81	6.30	3.21	9.97	2.31
fixed monitoring OT	0.74	0.10	0.75	0.12	0.74	0.15
free monitoring OT	0.74	0.13	0.74	0.14	0.73	0.13
fixed monitoring relative frequency	0.21	0.16	0.25	0.14	0.49	0.21
free monitoring relative frequency	0.22	0.22	0.20	0.13	0.40	0.16
fixed monitoring absolute frequency	0.48	0.17	0.44	0.20	0.42	0.20
free monitoring absolute frequency	0.39	0.20	0.40	0.18	0.57	0.15

Note: TBPM: time-based prospective memory; OT: ongoing task.

**Table 5 behavsci-14-00233-t005:** Regression analysis between age, monitoring conditions, relative frequency, absolute frequency and performance of TBPM.

	Predictor	*β*	*p*	*t*
TBPM	age	0.369 ***	<0.001	5.778
	monitoring conditions	0.033	0.629	0.484
	relative frequency	−0.001	0.992	−0.011
	absolute frequency	0.069	0.321	0.995

Note: *** *p* < 0.001.

## Data Availability

The raw data supporting the conclusions of this article will be made available by the authors on request.

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
