# Peer review of "The Effect of Time Monitoring on the Development of Time-Based Prospective Memory among Children Aged 7–11 Years Old"

_behavsci, 2024, doi:10.3390/bs14030233_

Round 1
Reviewer 1 Report
Comments and Suggestions for Authors
The manuscript by Wang et al investigates the development of time-based prospective memory (TBPM) in 7-, 9, and 11-year-old children. The authors employed time-point and time-period tasks and found age-related improvement in prospective memory. The main findings suggest that the ability to monitor time-dependent prospective memory tasks develops during childhood and depends on the type of the task and monitoring conditions. The authors raised the important and underexplored question of prospective memory development in childhood. They developed the experimental paradigm that allows to manipulate attentional resources during the TBPM task. Calculating both monitoring frequencies and the ability to prospectively monitor time to administer the stimuli in the timely manner provides the breadth of information regarding the factors predicting TBPM in children. While I believe that the manuscript will be of a great interest to developmental and cognitive psychologists, there are several points that need to be clarified prior the manuscript is published.
1. Please clarify the differences between the time-point and time-period tasks. From how it’s written, I understood that the time-period task was a dual task in which children performed the time-point task concurrently with the ongoing task that asked participants to discriminate between man-made and natural objects. However, the results suggest that the OT task was performed during both time-point and time-period tasks. It would be helpful if the authors reported the trial structure for both tasks and also described how children checked the clock while doing the OT task. I also suggest checking section 3.3.1. It looks like the fixed monitoring condition was described there twice.
2. More details on how the scoring system was developed would be greatly appreciated.
3. If the tasks require switching or concurrent performance (I could not understand it from the task description), it is possible that younger children have difficulty switching from one task to another, which may affect their PM task performance.
4. I am not sure that the results warrant the statement “These results suggested that age-related differences occur in the monitoring frequency of children aged 7-11 years under different monitoring conditions, i.e., young children are more inclined to improve retrieval scores by increasing the time monitoring frequency.” The main effect of age simply suggests that older (pre-adolescence) children checked the clock more frequently than younger children.
5. I suggest using a linear regression to examine if the interaction between the time monitoring frequency, age, monitoring conditions and frequency type can predict the TBPM score. The correlation analysis does not provide a necessary level of detail regarding the relationship between the TBPM score and the time monitoring frequency.
6. The manuscript will benefit from the linear regression analysis predicting the TBPM score from the monitoring frequency in Experiment 1 too.
7. Considering a large number of the analyses, I suggest placing some of them (e.g., the follow up analyses for age groups) into the table to make reading the results section a little bit easier.
8. p.13 “This finding shows that clear external cues are more helpful in improving TBPM performance among school-age children, supporting the results of previous studies” - Could you please clarify what external cues you are talking about?
Comments on the Quality of English LanguagePlease fix the typos. Also consider changing the terminology for the OT description. May be use natural - man-made instead of living-non-living. Technically speaking, apples are not alive.
Author Response
For research article(Manuscript ID: behavsci-2856304)
Response to Reviewer 1 Comments
Thank you very much for taking the time to review this manuscript. We express our sincere gratitude for your valuable and detailed suggestions. We have made careful reference to your review comments and revised them point by point. Below, we list all of the portions of the manuscript that have been revised based on your suggestions, and we have marked these modifications in red.
|
Comments 1: Please clarify the differences between the time-point and time-period tasks. From how it’s written, I understood that the time-period task was a dual task in which children performed the time-point task concurrently with the ongoing task that asked participants to discriminate between man-made and natural objects. However, the results suggest that the OT task was performed during both time-point and time-period tasks. It would be helpful if the authors reported the trial structure for both tasks and also described how children checked the clock while doing the OT task. I also suggest checking section 3.3.1. It looks like the fixed monitoring condition was described there twice. |
|
Response 1: Thanks so much for the valuable comments. We agree with your comments. (1)As for the first question in your comments, we have elaborated in detail on the differences between the point-in-time prospective memory task and the time-segment prospective memory task and modified the article. The specific modifications are as follows: First, the Time-point PM task, subjects were told to remember to press the “green” button every 2 minutes (corresponding to the “Enter” key) (For example, after the prospective memory instructions were introduced, that is, the PM block starts timing. The subjects need to make the key response within 1 minute before and after the target time point. For example, starting from 0:00, the subjects need to do the button reaction within 1 minute before and after the four time points of 2:00; 4:00; 6:00; 8:00. But the scores vary according to the time and location of the key pressed. The closer you are to the target time, the higher your score, and the further away you are, the lower your score). The child is also told that there is a clock in the upper right corner of the interface, which has been hidden since the prospective memory task and is responsible for timing. To remind children to monitor the passage of time, they can view the current clock by pressing the "yellow" button (the computer's "Ctrl" key); each time the "Ctrl" key is pressed, the clock displays for two seconds, then hides and continues counting. The clock continues to tick regardless of whether the Enter key is pressed or not. Second, in the Time period-based PM task, tell the subjects to press the "green" key (corresponding to the computer "Enter" key) after every 2 minutes while performing the task in progress. (For example, after we introduce the prospective memory instructions, that is, the PM block starts timing, the subjects need to make the key response within 1 minute after the target time. For example, starting from 0:00, the subjects need to do the button reaction within 1 minute after 2:00; 4:00; 6:00; 8:00. But the scores vary according to the time and location of the key pressed. The closer you are to the target time, the higher your score, and the further away you are, the lower your score). The child is also told that there is a clock in the upper right corner of the interface, which has been hidden since the prospective memory task and is responsible for timing.To remind children to monitor the passage of time, they can view the current clock by pressing the "yellow" button (the computer's "Ctrl" key); each time the "Ctrl" key is pressed, the clock displays for two seconds, then hides and continues counting. The clock continues to tick regardless of whether the Enter key is pressed or not. (see P5). (2)As for the second question, as you commented that we did describe the prospective memory task under fixed monitoring condition twice. We have changed the second description of the prospective memory task under fixed monitoring condition to the prospective memory task under free monitoring condition. The prospective memory task under free monitoring conditions is as follows: TBPM task under the free monitoring condition: Tell the subjects to press "green" button (corresponding to the computer "Enter" button) every 2 minutes (2:00; 4:00; 6:00; 8:00) while performing the ongoing task. Participants could press the “yellow” key (corresponding to the “Ctrl” key on the computer) to check the clock. Each time the “Ctrl” key is pressed, the clock is displayed for 2 seconds before it disappears. The clock is always ticking whether or not the “Ctrl” key is pressed. However, the number of times they looked at the clock was not limited, that is, they could monitor the time at any time as needed.(see P10). |
|
Comments 2: More details on how the scoring system was developed would be greatly appreciated. |
|
Response 2: Thanks so much for the valuable comments. Based on your comments, we will explain the scores of time-point and time-period prospective memory tasks in text and graphics. First, the scoring method of the time point prospective memory task: The key press reactions 1 minute before and after the target time are considered to have correctly performed the time point prospective memory task. However, the score differs depending on the specific time of the key press, with the score being higher when the key is pressed close to the target time. There are four prospective memory tasks, a full score of 12 points. For example, the first target time point is 2:00. The participants will be awarded 1 point for pressing the button between 1'-1'20" and 2'40"-3'; 2 points for pressing the button between 1'20"-1'40" and 2'20"-2'40'; and 3 points for pressing the button between 1'40"-2' and 2'-2'20". No points will be awarded for pressing the button at other time points. The total scores of the four time point test tasks will be accumulated as the participants' performance of time point-based prospective memory. See Figure 2 for details.( see P7).
Fig. 2 Schematic diagram of time-point prospective memory task scoring
Time period-based PM test: A button response within 1 minute after the target time was considered to indicate correct performance of the PM task. However, the score is different depending on the specific time period of the key press, with the score being higher when the key is pressed close to the target time.There are four prospective memory tasks, a full score of 12 points. For example, the first target time period is: within 1 minute after 2:00. The participants will be awarded 1 point for pressing the button between 2'40" and 3'; 2 points for pressing the button between 2'20" and 2'40'; and 3 points for pressing the button between 2' and 2'20". No points will be awarded for other time points. The total scores of the four time-based prospective memory tasks will be accumulated as the participants' performance of time-based prospective memory. See Figure 3 for details. (see P7).
Fig. 3 Schematic diagram of time-period prospective memory task scoring
|
|
Comments 3: If the tasks require switching or concurrent performance (I could not understand it from the task description), it is possible that younger children have difficulty switching from one task to another, which may affect their PM task performance. |
|
Response 3: Thanks so much for the valuable comments. According to your comments, we will refer to the task paradigm of temporal prospective memory in previous studies to explain that school-age children have the ability to complete this prospective memory task. It has been shown in previous studies that the task can be completed by the subjects at the school-age stage, for example, Ceci and Bronfenbrenner (1985) used this "dual task" paradigm to investigate the performance of TBPM in children aged 10-14 through situational simulation. Voigt et al. (2011) asked children aged 6-10 to complete a game task while remembering to fill up a car with gasoline; Ren et al. (2022) used the same paradigm to ask 8- to 12-year-old children to complete prospective memory tasks while doing ongoing tasks.Even younger children can perform this task, for example, Zhang (2013) examined the performance of temporal prospective memory in children aged 3-5 years old using a dual-task paradigm. This study used the classic “Dual Tasks Paradigm” proposed by Einstein and McDaniel (1990) to explore the developmental mechanism of temporal prospective memory of 7-11 group of school-age children from the perspective of time monitoring.(see P2-3). |
|
Comments 4: I am not sure that the results warrant the statement “These results suggested that age-related differences occur in the monitoring frequency of children aged 7-11 years under different monitoring conditions, i.e., young children are more inclined to improve retrieval scores by increasing the time monitoring frequency.” The main effect of age simply suggests that older (pre-adolescence) children checked the clock more frequently than younger children. |
|
Response 4: We agree with your comments. We have carefully sorted out the result part of monitoring frequency in Experiment 2, which is that we did not clearly express the meaning of the results. Therefore, we have revised the result part of monitoring frequency in Experiment 2. The main effect of monitoring frequency was significant, F(1, 175) = 90.59, p < .001, η2 = .34。The interaction of age and monitoring frequency was significant, F(2, 175) = 9.95, p < .001, η2 = .10. For 7-year-old children, the absolute frequency(M = 0.43, SD = 0.02)was significantly greater than the relative frequency(M = 0.22, SD = 0.02), t(59) = 7.45, p < .001; for 9-year-old children, the absolute frequency (M = 0.42, SD = 0.02) was also significantly greater than the relative frequency (M = 0.23, SD = 0.02),t(59) = 6.53, p < .001; however, for the 11-year-old group, the difference between relative frequency(M = 0.44, SD = 0.02) and absolute frequency (M = 0.49 SD = 0.02) was not significant, t(59) = 1.78, p = .08. There was a significant interaction among age, monitoring conditions, and monitoring frequency, F(2, 175) = 9.54, p < .001, η2 = .09. Further analysis revealed that, For 7-year-olds, under the fixed monitor condition, the absolute frequency(M = 0.48, SD = 0.17)was also significantly greater than the relative frequency(M = 0.21, SD = 0.16); under the free monitor condition, the absolute frequency(M = 0.39, SD = 0.20)was also significantly greater than the relative frequency(M = 0.22, SD = 0.14); For 9-year-olds, under the fixed monitor condition, the absolute frequency(M = 0.44, SD = 0.20)was also significantly greater than the relative frequency(M = 0.25, SD = 0.14); under the free monitor condition, the absolute frequency(M = 0.57, SD = 0.15)was also significantly greater than the relative frequency(M = 0.40, SD = 0.16). The results showed that the young children (7 and 9 years old) had poor monitoring strategy and could only improve prospective memory performance by frequently looking at the clock; Older (11 year old) children were better able to use time monitoring strategies when there was a limit on the number of monitoring sessions. (see P12-13). |
|
Comments 5: I suggest using a linear regression to examine if the interaction between the time monitoring frequency, age, monitoring conditions and frequency type can predict the TBPM score. The correlation analysis does not provide a necessary level of detail regarding the relationship between the TBPM score and the time monitoring frequency. |
|
Response 5: Thank you for your valuable comments. According to your suggestion, we have performed a linear regression analysis on the outcome part of the monitoring behavior in Experiment 2. The results have been revised in the article. Correlation between time monitoring behaviour and TBPM: Correlation analysis of the TBPM scores and the time monitoring frequency of the children aged 7-11 years revealed that the correlation coefficient between the relative time monitoring frequency and the TBPM score was 0.47 (p < 0.001). The higher the relative time monitoring frequency, the better the time-based prospective memory performance. Further, the performance of temporal prospective memory was taken as the dependent variable, and age, monitoring conditions and monitoring frequency type were taken as independent variables for stepwise regression analysis. The results showed that the regression coefficient of age on temporal prospective memory was significant, β = .40, t = 5.78, p < .001, Age has a significant positive predictive effect on time-based prospective memory performance. (see P13). |
|
Comments 6: The manuscript will benefit from the linear regression analysis predicting the TBPM score from the monitoring frequency in Experiment 1 too. |
|
Response 6: Thank you for your valuable comments, We agree with your comments. According to your suggestion, We have performed a linear regression analysis on the outcome part of the monitoring behavior in Experiment 1 based on your comments. Correlation between time monitoring behaviour and TBPM: Correlation analysis of the TBPM scores and the time monitoring frequency of the children aged 7-11 years revealed that the correlation coefficient between the relative time monitoring frequency and the TBPM score was 0.72 (p < 0.001). The higher the relative time monitoring frequency, the better the performance of time-based prospective memory; The correlation coefficient between the absolute time monitoring frequency and the TBPM score was 0.67 (p < 0.001). The higher the absolute time monitoring frequency, the better the temporal prospective memory performance. Further, the performance of temporal prospective memory was taken as dependent variable, and age, temporal prospective memory type and monitoring frequency type were taken as independent variables for stepwise regression analysis. The results show that relative monitoring frequency has significant regression coefficient on temporal prospective memory, β = .73, t = 11.37, p < .001, Relative monitoring frequency has significant positive predictive effect on time-based prospective memory performance. The margin of regression coefficient of age on time-based prospective memory is significant, β = .13, t = 1.97, p = .051. ( see P10). |
|
Comments 7: Considering a large number of the analyses, I suggest placing some of them (e.g., the follow up analyses for age groups) into the table to make reading the results section a little bit easier. |
|
Response 7: Thank you. We agree with your comments, According to your suggestion. We have added the corresponding table in the article, (see P7 and P11).
|
|
Comments 8: p.13 “This finding shows that clear external cues are more helpful in improving TBPM performance among school-age children, supporting the results of previous studies” - Could you please clarify what external cues you are talking about? |
|
Response 8: Thanks so much for the nice comments. The external information here refers to the time clue information.For example, in the introduction we introduced,If the TBPM time cue is a definite time point, then the time information it provides is relatively clear, and children can accurately predict when this TBPM cue will appear, whereby they can flexibly and efficiently allocate attentional resources based on explicit time information and activate target cues within the relevant time window, achieving the spontaneous retrieval of PM intentions(Maylor, 1990; Phillips et al., 2008; Rendell & Craik, 2000). If the TBPM time cue is a time range, then the time information it provides is rather vague, i.e., it provides relatively few or no environmental cues, and it cannot support the spontaneous retrieval and processing of intention(Rose et al., 2010). In other words, under relatively fuzzy time cues, individuals may be less effective at allocating attentional resources, thus affecting their execution of TBPM tasks(Einstein et al., 2005). |
|
Comments on the Quality of English Language: : Please fix the typos. Also consider changing the terminology for the OT description. May be use natural - man-made instead of living-non-living. Technically speaking, apples are not alive.
|
|
Response : Thanks so much for the nice comments. We agree with your comments. We made a mistake about whether apples have life. The original experimental material used was apple trees, but we miswrote apples.“apple ”revised them to: “apple trees”. (see P4). The terms about OT description have been modified in the whole text. |
|
Additional clarifications |
|
We have revised other parts of the article according to the modification suggestions.
|
|
Finally, thank you very much for your valuable and specific comments. We have learned much from the suggestions. If there are any questions, please let us know. We would love to further revise the manuscript. Thank you. |
|
Best regards, Authors |
Reviewer 2 Report
Comments and Suggestions for Authors
This manuscript reports two experiments that showed that the developmental trend in time-based prospective memory (TBPM) was modified by the type of prospective memory (time-point vs. time-period) and time monitoring condition (free vs. mixed monitoring). The studies address an interesting question, and the results are straightforward. However, there are some inconsistencies between the conclusions and the results, and there are some missing pieces in the logical flow that prevent me from recommending this paper for publication.
1. The authors claimed in the general discussion that “this study showed that the development of time point PM ability among school-age children is significantly greater than that of their time period PM ability” and in the conclusion that “The development of time point PM ability occurs *earlier* than that of time period PM ability.” I don’t see why the results would point to such conclusions. The ANOVA results for PM performance showed that for time point PM, the trend is 7 year old = 9 year old < 11 year old, while the trend for time period PM is 7 year old < 9 year old = 11 year old. Isn’t that both PM performance increase with age, while the age-related increase in time point PM occurred *later* than time period PM?
2. The authors stated that the study hypothesized that the development of time point PM ability occurs significantly earlier than that of time period PM ability. They also stated that “In comparison, the performance of time point PM is less affected by age.” However, I see not sufficient support laid out for these claims. I suggest the authors elaborate on the rationales behind these hypothesis and claim.
3. A main motivation for the current study is to address the factors that may lead to the mixed findings in the literature and the authors identify two factors: type of prospective memory and time monitoring condition. To help readers better understand why the authors focus on the two specific factors, I suggest adding a table listing out the age range, amount of time monitoring, and the focal TBPM task in previous studies and their corresponding findings.
4. At the bottom of p.4, the instructions for time-point PM and time-duration PM seem very similar, please clarify their differences.
5. There are two citation and reference styles used in parallel in the paper, which is confusing.
Minor points:
1. In line 179, the participants are said to be “the patients”
Comments on the Quality of English LanguageThere are many grammar errors, typos, and typesetting errors (e.g., lines 72, 76, 204-206, 208, etc.). I recommend more careful proofreading and language polishing.
Author Response
For research article(Manuscript ID: behavsci-2856304)
Response to Reviewer 2 Comments
Thank you very much for taking the time to review this manuscript. We express our sincere gratitude for your valuable and detailed suggestions. We have made careful reference to your review comments and revised them item by item. Below, we list all of the portions of the manuscript that have been revised based on your suggestions, and we have marked these modifications in red.
|
Comments 1: The authors claimed in the general discussion that “this study showed that the development of time point PM ability among school-age children is significantly greater than that of their time period PM ability” and in the conclusion that “The development of time point PM ability occurs *earlier* than that of time period PM ability.” I don’t see why the results would point to such conclusions. The ANOVA results for PM performance showed that for time point PM, the trend is 7 year old = 9 year old < 11 year old, while the trend for time period PM is 7 year old < 9 year old = 11 year old. Isn’t that both PM performance increase with age, while the age-related increase in time point PM occurred *later* than time period PM? |
|
Response 1: Thank you for your valuable comments, and we agree with your comments. (1) As for your first comment, in the general discussion, we mentioned that "this study showed that the development of time point PM ability among school-age children is significantly greater than that of their time period PM ability". This sentence is indeed not clear enough in our analysis and discussion of the results. As your comment, "this study showed that the development of time point PM ability among school-age children is significantly greater than that of their time period PM ability" should be changed to "This study shows that the performance of time point prospective memory of school-age children is significantly better than that of time period prospective memory ; and the performance of time point and time period prospective memory of school-age children continues to develop with age. This shows that clear external time cues are more helpful to improve the performance of time point prospective memory of school-age children, which verifies the previous research results (Chen Youzhen et al., 2010, 2014). At the same time, the two types of prospective memory abilities of school-age children do not develop in sync, and the time period prospective memory ability mainly concentrates on the stage of 7-9 years old children. The time point prospective memory ability is mainly concentrated in the stage of 9-11 years old children. The development of time period prospective memory takes precedence over the development of time point prospective memory”. (see P15). (2) As for your second comment, in the conclusion, we mentioned that "the development of time point PM ability is *earlier* than that of time period PM ability". It is our mistake, and we should change "the time-based prospective memory ability of school-age children continues to develop with the increase of age. But the performance of the two types of prospective memory doesn't develop in sync, the development of school-age children's prospective memory ability of time point is later than that of time period. The development of time point prospective memory ability mainly occurs in the 9-11 years old group. The development of time period prospective memory ability mainly occurs in the 7-9 year-old group".(see P17). |
|
Comments 2: The authors stated that the study hypothesized that the development of time point PM ability occurs significantly earlier than that of time period PM ability. They also stated that “In comparison, the performance of time point PM is less affected by age.” However, I see not sufficient support laid out for these claims. I suggest the authors elaborate on the rationales behind these hypothesis and claim.
|
|
Response 2: Thank you for your comments. We agree with your comments. (1) As for the hypothesis part "The development of different temporal prospective memory abilities of school-age children is not synchronous, and the development of temporal prospective memory ability is significantly earlier than that of temporal prospective memory ability". We did not express clearly. We should change "The development of different time-based prospective memory abilities of school-age children is not synchronous, and the development of time point prospective memory ability is significantly earlier than that of time period prospective memory ability" to "The performance of time-based prospective memory of school-age children improves with age, but the development of two types of time-based prospective memory abilities is not synchronous; and the performance of time period prospective memory is better than that of time point prospective memory".(see P4). (2) As for the introduction part mentioned "In comparison, the performance of time point PM is less affected by age". What we really want to express is that "... In other words, under the condition of relatively fuzzy time cues, the effectiveness of individual allocation of attention resources may be poor, which will affect the execution of temporal prospective memory tasks(Einstein & Mcdaniel, 2005). This difference is similar to the difference between focal event cues (cues related to the cognitive processing involved in the ongoing task) and non-focal event cues (cues unrelated to the cognitive processing involved in the ongoing task) in event prospective memory. Focused event cue tasks involve relatively spontaneous extraction processes and are less affected by age (Einstein & Mcdaniel, 2005), while non-focused event cue tasks involve relatively difficult and strategic cognitive processes and are more affected by age (Kliegel et al., 2008; Mcdaniel & Einstein, 2007).(see P3-4).
|
|
Comments 3: A main motivation for the current study is to address the factors that may lead to the mixed findings in the literature and the authors identify two factors: type of prospective memory and time monitoring condition. To help readers better understand why the authors focus on the two specific factors, I suggest adding a table listing out the age range, amount of time monitoring, and the focal TBPM task in previous studies and their corresponding findings. |
|
Response 3: Yes, Thank you so much for your valuable comments. We agree with your comments. We have created a table based on your comments, (see P4-5). |
|
Comments 4: At the bottom of p.4, the instructions for time-point PM and time-duration PM seem very similar, please clarify their differences. |
|
Response 4: Thanks so much for the valuable comments. We agree with your comments. (1)As for the first question in your comments, we have elaborated in detail on the differences between the point-in-time prospective memory task and the time-segment prospective memory task and modified the article. The specific modifications are as follows: First, the Time-point PM task, subjects were told to remember to press the “green” button every 2 minutes (corresponding to the “Enter” key) (For example, after the prospective memory instructions were introduced, that is, the PM block starts timing. The subjects need to make the key response within 1 minute before and after the target time point. For example, starting from 0:00, the subjects need to do the button reaction within 1 minute before and after the four time points of 2:00; 4:00; 6:00; 8:00. But the scores vary according to the time and location of the key pressed. The closer you are to the target time, the higher your score, and the further away you are, the lower your score). The child is also told that there is a clock in the upper right corner of the interface, which has been hidden since the prospective memory task and is responsible for timing. To remind children to monitor the passage of time, they can view the current clock by pressing the "yellow" button (the computer's "Ctrl" key); each time the "Ctrl" key is pressed, the clock displays for two seconds, then hides and continues counting. The clock continues to tick regardless of whether the Enter key is pressed or not. Second, in the Time period-based PM task, tell the subjects to press the "green" key (corresponding to the computer "Enter" key) after every 2 minutes while performing the task in progress. (For example, after we introduce the prospective memory instructions, that is, the PM block starts timing, the subjects need to make the key response within 1 minute after the target time. For example, starting from 0:00, the subjects need to do the button reaction within 1 minute after 2:00; 4:00; 6:00; 8:00. But the scores vary according to the time and location of the key pressed. The closer you are to the target time, the higher your score, and the further away you are, the lower your score). The child is also told that there is a clock in the upper right corner of the interface, which has been hidden since the prospective memory task and is responsible for timing.To remind children to monitor the passage of time, they can view the current clock by pressing the "yellow" button (the computer's "Ctrl" key); each time the "Ctrl" key is pressed, the clock displays for two seconds, then hides and continues counting. The clock continues to tick regardless of whether the Enter key is pressed or not. (see P5). |
|
Comments 5: There are two citation and reference styles used in parallel in the paper, which is confusing. |
|
Response 5: Yes, we agree with the comments. We have retained the correct format of the citation required by this journal and have modified it in the paper.
|
|
Minor points: In line 179, the participants are said to be “the patients”.
|
|
Response :Thanks so much for the specific comments. we agree with the comments. This is an error we made and we have corrected it. Please change "the patients" to "the subjects". (seeP5). |
|
Comments on the Quality of English Language: There are many grammar errors, typos, and typesetting errors (e.g., lines 72, 76, 204-206, 208, etc.). I recommend more careful proofreading and language polishing. |
|
Response : Thanks so much for the specific comments. We have carefully proofread and revised the paper.
|
|
Additional clarifications We have revised other parts of the article according to the modification suggestions.
|
|
Finally, thank you very much for your valuable and detailed comments. We have learned much from the suggestions. If there are any questions, please let us know. We would love to further revise the manuscript. Thank you. |
|
Best regards, Authors |
Round 2
Reviewer 1 Report
Comments and Suggestions for Authors
The authors addressed all my comments.
Reviewer 2 Report
Comments and Suggestions for Authors
The authors have addressed all my concerns, and I am now able to recommend the manuscript for publication.
Comments on the Quality of English LanguageThere are still some typos and grammar errors, so further rounds of editing of English language will be needed.